# Validation of the Perception Neuron system for full-body motion capture

**Corliss Zhi Yi Choo**, **Jia Yi Chow***, **John Komar**

Physical Education and Sports Science, National Institute of Education, Nanyang Technological University, Singapore, Singapore

* jiayi.chow@nie.edu.sg

## Abstract

Recent advancements in Inertial Measurement Units (IMUs) offers the possibility of its use as a cost effective and portable alternative to traditional optoelectronic motion capture systems in analyzing biomechanical performance. One such commercially available IMU is the Perception Neuron motion capture system (PNS). The accuracy of the PNS had been tested and was reported to be a valid method for assessing the upper body range of motion to within 5° RMSE. However, testing of the PNS was limited to upper body motion involving functional movement within a single plane. Therefore, the purpose of this study is to further validate the Perception Neuron system with reference to a conventional optoelectronic motion capture system (VICON) through the use of dynamic movements (e.g., walking, jogging and a multi-articular sports movement with object manipulation) and to determine its feasibility through full-body kinematic analysis. Validation was evaluated using Pearson's R correlation, RMSE and Bland-Altman estimates. Present findings suggest that the PNS performed well against the VICON motion analysis system with most joint angles reporting a RMSE of < 4° and strong average Pearson's R correlation of 0.85, with the exception of the shoulder abduction/adduction where RMSE was larger and Pearson's R correlation at a moderate level. Bland-Altman analysis revealed that most joint angles across the different movements had a mean bias of less than 10°, except for the shoulder abduction/adduction and elbow flexion/extension measurements. It was concluded that the PNS may not be the best substitute for traditional motion analysis technology if there is a need to replicate raw joint angles. However, there was adequate sensitivity to measure changes in joint angles and would be suitable when normalized joint angles are compared and the focus of analysis is to identify changes in movement patterns.

**Data Availability Statement:** All relevant data are within the manuscript and its Supporting Information files. Any other data are available in the NIE data repository through this link: https://doi.org/10.25340/R4/IZKYZR.

## Introduction

Traditionally, motion capture has been collected through the use of high-speed optical systems which allows for dynamic movements to be captured and analysed [1, 2]. These systems serve as a reliable and suitable methods to measure complex movements and are often considered as the gold standard in motion capture, providing an estimated accuracy (RMS error) of less than 1.00° and 1.50° for static and dynamic measurements respectively [2–4]. An example of a

**Funding:** The author(s) received no specific funding for this work.

**Competing interests:** The authors have declared that no competing interests exist.

high-speed optical system is the VICON motion analysis system (Oxford Metrics Group Ltd., Oxford, UK). For such systems, passive reflective markers are placed on specific body landmarks which reflect light back into the sensors of the cameras and the system computes the 3D positions of the makers within the environment using the triangulation method [1, 2, 4].

3D optoelectronic motion capture systems have improved significantly over the years due to higher camera resolution, better calibration techniques, and improvements in manufacturer tracking software which allows for more accurate marker tracking and greater post processing capabilities [1, 5]. With these changes in technology, consistency in accuracy by such systems have showed a nearly three-fold improvements in agreement over the last 20 years with no systems exceeding an error of 1.0mm [5]. Due to their validity and reliability as compared to other motion capture systems [2], optoelectronic motion capture systems have been extensively used in a variety of field, ranging from functional movement tasks [6, 7] to sports performance [8, 9] and injury prevention [10, 11], thus contributing meaningful information and advancements to the field of biomechanics over the years [1].

Although optical based systems have been commonly used for motion analysis, they are still expensive, have somewhat limited portability and typically require movement to be recorded within the confines of a laboratory environment [4, 12, 13]. Set-up and calibration also involves a lengthy preparation process [14]. Furthermore during data collection, markers have to be in line-of-sight of the cameras as any occlusion or alterations to camera positions will affect the reliability of the data [2]. Thus even though optical based systems are considered as the gold standard with consistently accurate measurements, the desire by researchers to collect accurate and reliable kinematic in applied situations for research and for the routine analysis of movements within normal training environments without being obstructed by environmental restrictions and lengthy processing time has increasingly driven researchers to search for alternatives [1, 15].

One such alternative is the inertial measurement unit (IMU). IMUs are devices that comprise of a gyroscope, an accelerometer and magnetometer and measures the motion of the user through embedded data fusion, human body dynamics and physical engine algorithms which allows for the estimation of kinematic data of body segments within a 3D space to be obtained [3, 16]. These IMUs allow users to capture and record real-time motion with a fast set-up process and minimal space restrictions thus increasing the portability and allowing movement capture to be conducted within real-life environments [16]. Hence, IMUs could potentially serve as a solution to the limitations that optical based motion capture systems face.

Importantly, recent advancements in IMUs offers the possibility of its use as a cost effective, portable alternative to analyzing biomechanical performance in an efficient way [3, 17, 18]. In general a combination of higher sampling rate, better sensor fusion algorithms and lower drift have resulted in increased accuracy of the data collected by these sensors [19]. As such IMUs have been used in various applications in sports, such as technique identification and performance, tracking athletes' performance and for injury screening [2, 14, 20]. One example of technique identification and performance is a study conducted by Bosch, Shoaib [21], where researchers were able to distinguish between experienced and novice rowers by simply placing sensors on the lower leg, the lower back and upper back and analyzing consistency in postural angles and timing of the strokes. With respect to tracking athletes' performance, Shepherd, Giblin [22] used IMUs effectively to monitor shooting kinematics and coordination patterns of netball athletes. This provided coaches and athletes with actionable insights to develop practices to achieve more consistent forearm angle at ball release and increase their chances of scoring during a game. IMUs have also been used for injury screening and prevention by evaluating movements and specifically to identify the presence of abnormal movements during performance [23]. They allow scientists to quantitatively monitor and detect movement

kinematics or volumes of training which has been linked to overuse injuries and hence allow them to issue preventive warnings [23–25]. As seen from the studies above, information obtained from these IMUs could be utilized by sports scientists and professional coaches in a myriad of ways and within ecological contexts [17, 18, 26]. Furthermore with a combination of lower costs, greater ease of use, increased portability for use in the field, it could eventually drive the use of these IMUs in sports not only exclusively for elite athletes but for a much wider target audience [14, 27]. However, accuracy of the IMUs depends on the specific sensors used, the software algorithms of the tested IMU, and the measured joint and task that is being performed [19]. Therefore it is necessary that specific validation of each new IMU system is completed before any IMU can be used routinely in assessments [19].

One such commercially available IMU is the Perception Neuron motion capture system (PNS) (Perception Neuron, Noitom, Miami, FL, USA). It is an adaptable and affordable motion capture system offering user-friendly technology that was developed to analyse movements for various industries from film makers to game developers, biomechanics researchers and sports and medical analysts. Some of the benefits of the PNS includes its ease of use since the sensors are interchangeable and also the portability of the system due to the ability to rely on a portable battery for its energy needs. In addition, the PNS has the capacity to record the data locally on a memory card or transfer the data through WiFi instead of a wired connection.

Currently, validation of the PNS is still limited. One such validation study was conducted by Sers, Forrester [12], where the accuracy of joint angles measurements acquired by the PNS was referenced against a gold standard optoelectronic systems (VICON). Movements analysed included neck flexion/extension, neck lateral flexion, neck rotation, torso flexion/extension, torso later flexion, torso rotation and shoulder abduction. Participants were tasked to perform each movement twice and at self-selected fast and slow speeds. Results suggests that the PNS is a valid method to assess the majority of the upper body range of motion during static movements with systematic and random bias for majority of the range of motion differences being below 4.5˚, with an exception for the knee extension where values were 6.1–9.0˚. However Sers, Forrester [12] only analysed the validity of the PNS using only postural angular kinematics of the upper body.

Therefore, the purpose of the study was to further validate the PNS with reference to a conventional optoelectronic motion capture system for full-body human movement during more dynamic movements and to determine its feasibility in a comprehensive full body kinematic analysis. Specifically, performance of the PNS was assessed using kinematic measures (e.g., joint angles) in comparison to the VICON motion analysis system using a series of full-body human movements. Walking, jogging, and floorball wrist shots were used as a proposed task vehicle as these activities are predicated on its feature as a discrete multi-articular task that involves the coordination of multiple body parts, joints and limb segments with object manipulation. The wrist shot was specifically chosen out of the many shots available within floorball as it is the most common shot used by players to score a goal and it is executed at a fast pace. The inclusion of these movements allows the testing of the PNS' ability to measure kinematic data during dynamic and fast paced tasks, which was previously not examined in the study undertaken by Sers, Forrester [12]. In addition, results from this study could potentially provide evidence for researchers that the PNS could be a viable tool to collect motion capture data in a more representative environment outside of a lab setting. This is especially so when it is not feasible to set up an optical based system. For example, in situations where there is a need to analyse movement over a larger area, if an optical based system is used, a greater number of cameras would be required to cover this increase in volume. In this case, the PNS could be

used to overcome the limitations of the lab space and the number of cameras needed whilst still providing valid, reliable and objective data.

It is hypothesized that measurements from the PNS would be comparable to the measurements from a traditional optoelectronic motion capture analysis system (VICON), thus validating the use of the PNS as an alternative for full-body human motion capture. By establishing the validity of the PNS, it ensures that the system can adequately measure full-body movement. The PNS could then potentially be used as an alternative to the current optical motion systems during sports biomechanical applications. This could help mitigate the restrictions of the traditional 3D optical motion system which are expensive and better allow collection of data within real-world settings outside the restricted confines of a laboratory.

## Methods

### Participants

Ten participants (7 males and 3 females) with a mean age of 26 ± 2 years, mean height of 1.70 ± 0.09 m and mean weight of 60.48 ± 11.91 kg participated in this study. This study was approved by the University's Institutional Review Board (IRB-2019-03-005) and written consent was obtained prior to the start of any data collection. Exclusion criterion was musculoskeletal injury, illness or disease for the past 6 months prior to the time of testing, mainly to ensure safety for the participants.

### Instrumentation

Movement data was collected simultaneously with the PNS and an eight-camera VICON MX40 optical motion capture system recording at 120Hz each. Participants' movement were also captured and recorded via two digital video cameras (CASIO EXILIM EX100, Tokyo, Japan) from two planes of observation (frontal and sagittal). The VICON optoelectronic motion analysis system was chosen as it is a widely used system and is considered as the current laboratory gold-standard with a high accuracy during dynamic trials [12, 19, 28].

Thirty-three reflective markers (19-mm) were placed on anatomical landmarks based on the Full-body modeling with Plug-in Gait (Vicon Peak, Oxford, UK). There was one adjustment made to the marker set, this was to the T10 marker. The T10 marker was shifted to the same level as the sternum for easier location purposes. The PNS consisted of 17 IMUs placed at specific landmarks in accordance to PNS guidelines as shown in Fig 1. The PNS suit was connected to a computer through a USB connection and the Axis Neuron software (AXIS Neuron, Noitom, Miami, Florida, USA) was used to view the data in real-time.

Calibration of both systems was also completed prior to each movement recording. The PNS was calibrated according to the manufacturer's instructions using a four-step calibration process (see Table 1) and was zeroed after five repetitions of each task to remove any errors. The VICON motion analysis system was calibrated using the T-pose.

### Protocol

All participants underwent one test session. Anthropometric measurements including height, weight, leg length, knee width, ankle width, elbow width and hand thickness was recorded from every participant at the start of the session. Participants were then suited up in the PNS 17-neuron full body suit and the 39 19-mm retroreflective markers for the VICON system were placed on anatomical landmarks according to the Plug-in-Gait full-body model (VICON Motion Systems, Oxford Metrics Group Ltd) and held in place using medical grade double-sided adhesive tape as seen in Fig 2.

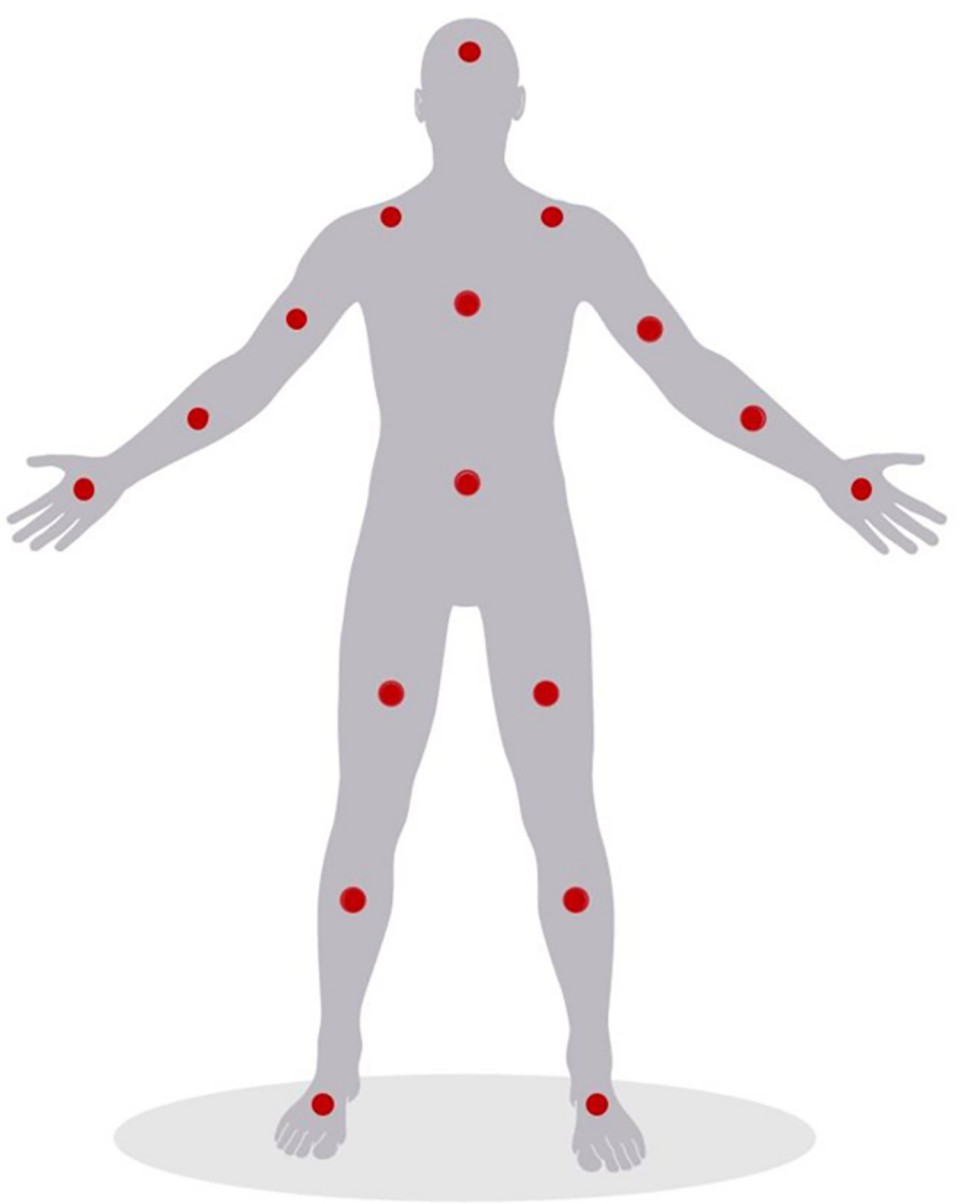

**Fig 1. Perception Neuron IMU sensor configuration.**

**Table 1. Perception Neuron 4-step calibration.**

| Posture calibration | Details |
| --- | --- |
| 1. Steady pose | Participant to be seated and to remain as still as possible |
| 2. A-pose | Participant to place palms down on the side of the thighs and keep feet parallel |
| 3. T-pose | Participants to have shoulders abducted by 90˚ with the palms facing the floor |
| 4. S-pose | Participants to bend knees approximately 45˚ and place arms in front and position them parallel to the floor |

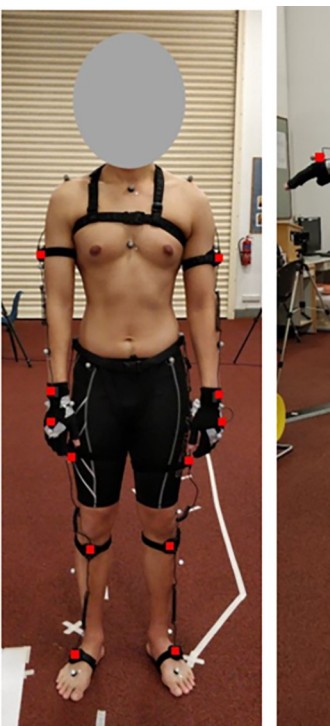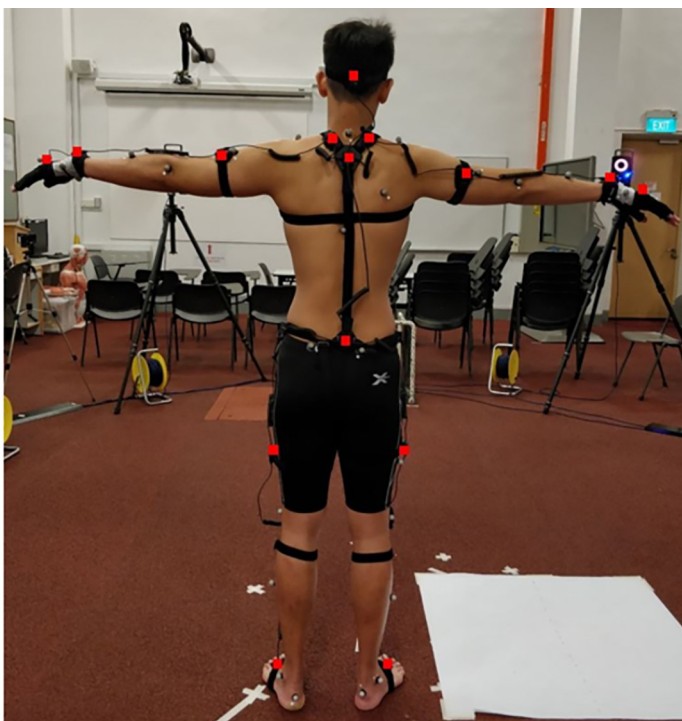

**Fig 2. Participant set-up with both PNS and VICON markers front and back.** The red squares indicates the PNS sensor positions.

Participants were required to perform a series of tasks. Description of these can be found in Table 2. Participants performed 10 repetitions for each task. During the performance of the tasks, care was taken to ensure that participants' movement were not obstructed in any way as a result of the reflective markers and the PNS. Compression garments (2XU Compression Shorts, Melbourne, Australia) were used during the data collection to minimize the marker movements whilst still allowing participants to be covered. Compression garments were chosen as it had been previously reported that markers placed over compression garments demonstrate low variance in movement compared to the markers placed on the skin [29].

**Table 2. Description of tasks to be performed.**

| Task | Description |
| --- | --- |
| Stationary walk | Participants to swing their arms, bring their knees up and feet back down to the same point where they took off from at a self-determined walking pace to mimic the walking movement. Participants performed the task whilst remaining within a 90 x 90 cm demarcated spot on the lab floor |
| Distance walk | Participants to walk along a 3m line on the lab floor at a self-determined walking pace |
| Stationary jog | Participants to swing their arms, bring their knees up and feet back down to the same point where they took off from at a self-determined jogging pace to mimic the jogging movement. Participants performed the task whilst remaining within a 90 x 90 cm demarcated spot on the lab floor |
| Distance jog | Participants to walk along a 3m line on the lab floor at a self-determined jogging pace |
| Stationary floorball wrist shot | Participants to stand 4m in front of an open goal and take shots towards the goal |
| Moving wrist shot | Participants to stand 7m in front of an open goal, dribble the ball along a 3m line and take a shot towards goal once they are 4m in front of the goal |

## Data analysis

Data collected from the PNS were exported as a processed bvh file. Displacement of the VICON markers in 3D space were reconstructed and manually labelled for all trials using VICON Nexus 2.9.0. Joint angle data from both systems were calculated from the 3D coordinates using a custom script programmed using MATLAB (R2018b, Mathworks Inc., Natick, MA USA) which computes both data on the same basis in order to ensure that data from both systems are comparable. Elbow flexion/extension, shoulder flexion/extension, shoulder abduction/adduction, hip flexion/extension, hip abduction/adduction, knee flexion/extension and ankle flexion/extension angles were calculated from both systems and comparisons between measurements obtained from the two systems were made. These flexion angles were adapted based on the study of Lee, Chow [30] and Lazzaeri, Kayser [8]. Subsequently, joint angles from both devices were smoothed using the "sgolayfilt" function with a polynomial function of degree 3 in MATLAB [31]. The Savitzky-Golay filter is based on the principle of local least squares fitting of a polynomial approximation [32, 33]. This filter is effective at preserving the width and height of the waveform [32, 33]. In addition to using the same recording speed, all trials were also time normalized to 100 points to allow for simultaneous comparisons.

To compare the joint angles from the VICON and PNS, Pearson's correlation and RMSE were evaluated for all movements on the raw joint angles. Pearson correlation was used to test the linear relationship between the joint angle measurement from PNS and VICON system. Correlation values were interpreted as such (in either direction positive or negative direction): 0–0.3 represents a weak relationship, 0.3–0.7 represents a moderate relationship and 0.7–1.0 represents a strong relationship [34]. Correlations were calculated between the VICON and PNS per joint, per activity and per subject. The standard deviation, mean and p-value reported were based on the average of these correlations. All statistical analyses were completed using MATLAB.

RMSE was used to compare the two methods as a measure of fit [35],

$$RMSE = \sqrt{\frac{\sum (X(t) - Y(t))^2}{n}}$$

Where X(t) is the value from the PNS suit, Y(t) is the observed value from VICON and $n$ is the total number of time points.

Group averages and SD were presented for Pearson correlation and RMSE.

The construction of Bland-Altman plots [36, 37] was another way to determine the relationship between the two measurement systems. Correlation exploration between the error and the mean value was included to examine for the existence of heteroscedasticity [38]. Mean bias and limits of agreement (LOA) were calculated according to Bland and Altman [37] for comparisons on raw joint angles to check for systematic errors. LOA was calculated as such,

$$LOA = \bar{d} \pm 1.96SD$$

where $\bar{d}$ refers to the mean difference and $SD$ refers to the standard deviation.

LOA (%) was also used to check for the agreement between joint angle calculations from the two measurement systems. LOA (%) was defined as the percentage of readings that lie within the limits of agreement. This was calculated for both raw and normalized joint angles. Normalization of joint angle data were conducted using z-score transformation as such,

$$z = \frac{x - \mu}{\sigma}$$

**Table 3. Pearson's correlation comparison between PNS and VICON.**

| Joint Angle | Stationary walk | Distance walk | Stationary jog | Distance jog | Stationary wrist shot | Distance wrist shot |
|---|---|---|---|---|---|---|
| Elbow (flex/ext) | 0.96 | 0.91 | 0.85 | 0.94 | 0.84 | 0.85 |
| SD | 0.06 | 0.17 | 0.31 | 0.08 | 0.25 | 0.17 |
| p-value | < 0.001 | 0.0023 | 0.026 | < 0.001 | 0.014 | < 0.001 |
| Shoulder (flex/ext) | 0.80 | 0.59 | 0.82 | 0.88 | 0.76 | 0.91 |
| SD | 0.20 | 0.33 | 0.29 | 0.18 | 0.33 | 0.15 |
| p-value | 0.0038 | 0.047 | 0.040 | 0.004 | 0.032 | 0.0053 |
| Shoulder (abd/add) | 0.77 | 0.66 | 0.79 | 0.87 | 0.58 | 0.50 |
| SD | 0.21 | 0.43 | 0.29 | 0.14 | 0.50 | 0.46 |
| p-value | < 0.001 | 0.03 | 0.04 | < 0.001 | 0.02 | 0.04 |
| Hip (flex/ext) | 0.95 | 0.82 | 0.73 | 0.90 | 0.84 | 0.82 |
| SD | 0.06 | 0.22 | 0.35 | 0.08 | 0.28 | 0.25 |
| p-value | < 0.001 | 0.0016 | 0.039 | < 0.001 | 0.0077 | 0.016 |
| Hip (abd/add) | 0.63 | 0.84 | 0.71 | 0.74 | 0.93 | 0.89 |
| SD | 0.40 | 0.17 | 0.42 | 0.26 | 0.13 | 0.17 |
| p-value | 0.0325 | 0.0039 | 0.037 | 0.012 | 0.0072 | 0.0024 |
| Knee (flex/ext) | 0.99 | 0.99 | 0.88 | 0.99 | 0.80 | 0.96 |
| SD | 0.00 | 0.02 | 0.30 | 0.01 | 0.37 | 0.06 |
| p-value | < 0.001 | < 0.001 | 0.037 | < 0.001 | 0.011 | < 0.001 |
| Ankle (flex/ext) | 0.59 | 0.82 | 0.45 | 0.73 | 0.55 | 0.78 |
| SD | 0.53 | 0.22 | 0.54 | 0.36 | 0.50 | 0.19 |
| p-value | 0.0082 | 0.018 | 0.15 | 0.024 | 0.056 | 0.0034 |

where $x$ refers to the data point value, µ refers to the mean and $\sigma$ represents the standard deviation.

## Results

Table 3 contains the Pearson's R correlation between the PNS and VICON joint angle calculations. Most correlation values were statistically significant, with the exception of ankle flexion/extension during stationary jog and stationary wrist shot. Most values showed strong positive correlation with the exception of hip abduction/adduction during stationary walk at 0.63 ± 0.40, shoulder flexion/extension during distance walk with a value of 0.59 ± 0.33, shoulder abduction/adduction during stationary wrist shot, distance wrist shot and during distance walk with a correlation of 0.58 ± 0.50, 0.50 ± 0.46 and 0.66 ± 0.43 respectively, and ankle flexion/extension during stationary walk, stationary jog and stationary wrist shot with a correlation of 0.59 ± 0.53, 0.45 ± 0.54 and 0.55 ± 0.50 respectively. Correlation was strong for knee flexion/extension across all movements with correlation ranging between 0.80 and 0.99.

Table 4 displays the RMSE comparisons between the PNS and VICON joint angle calculations on the raw values. RMSE values for raw values were generally below 4° for all joint angles and across all movements except for shoulder abduction/adduction for all movements and for ankle flexion/extension during stationary jog and distance jog.

Tables 5 and 6 includes the mean bias and limits of agreement (LOA) for each joint angle and across all movements. The ankle flexion/extension showed the largest mean bias of 22.74° during stationary walk. Whereas the hip flexion/extension had the least differences as compared to the other joint angles with a mean bias of 0.56° during stationary walk. There was a trend for joint angles to be overestimated using the PNS as compared to the VICON across movements. Only the angles for shoulder flexion/extension and shoulder abduction/adduction

**Table 4. RMSE comparison between PNS and VICON for raw values for all movements.**

| Joint Angle | Stationary walk | Distance walk | Stationary jog | Distance jog | Stationary wrist shot | Distance wrist shot |
|---|---|---|---|---|---|---|
| Elbow (flex/ext) | 3.40 | 2.04 | 3.89 | 1.92 | 2.81 | 3.20 |
| SD | 2.15 | 1.48 | 2.96 | 1.00 | 2.18 | 1.75 |
| Shoulder (flex/ext) | 1.90 | 1.12 | 1.94 | 1.78 | 2.23 | 1.99 |
| SD | 0.80 | 0.65 | 1.53 | 1.16 | 1.97 | 1.12 |
| Shoulder (abd/add) | 7.14 | 5.36 | 5.97 | 5.70 | 11.85 | 15.15 |
| SD | 2.97 | 3.16 | 3.80 | 2.57 | 10.24 | 9.32 |
| Hip (flex/ext) | 2.78 | 2.63 | 2.91 | 2.76 | 1.83 | 3.58 |
| SD | 1.61 | 1.19 | 1.51 | 1.23 | 1.29 | 2.68 |
| Hip (abd/add) | 1.50 | 1.47 | 1.32 | 1.80 | 1.00 | 1.67 |
| SD | 0.95 | 1.29 | 0.85 | 0.95 | 0.62 | 0.86 |
| Knee (flex/ext) | 1.48 | 2.56 | 5.26 | 2.19 | 1.12 | 3.35 |
| SD | 0.56 | 1.16 | 9.13 | 1.27 | 0.81 | 2.23 |
| Ankle (flex/ext) | 2.77 | 3.11 | 4.33 | 4.47 | 1.21 | 3.83 |
| SD | 1.20 | 1.37 | 3.07 | 1.94 | 1.34 | 1.65 |

was underestimated. Existence of heteroscedasticity was found for most joint angles and across all movements. An example of a Bland-Altman plot for hip flexion/extension during distance wrist shot can be found in Fig 3. All the Bland-Altman plots for all movements and joint angles can be found in S1 Appendix.

Table 7 shows the LOA (%) between the PNS and VICON joint angle calculations for both raw and normalized values. LOA (%) for raw values ranged from 92.61% to 97.48% across all

**Table 5. Limits of agreement and mean bias for all stationary movements.**

| | Stationary walk | | | Stationary jog | | | Stationary wrist shot | | |
|---|---|---|---|---|---|---|---|---|---|
| Joint Angle | Lower limit | Mean bias | Upper limit | Lower limit | Mean bias | Upper limit | Lower limit | Mean bias | Upper limit |
| Elbow (flex/ext) | -42.55 | -16.20 | 10.15 | -68.01 | -15.06 | 37.90 | -40.53 | -20.34 | -0.15 |
| correlation (heteroscedasticity) | $p < 0.001$ | $r = -0.62$ | | $p < 0.001$ | $r = -0.52$ | | $p < 0.001$ | $r = -0.17$ | |
| coefficient of determination | $r^2 = 0.38$ | | | $r^2 = 0.27$ | | | $r^2 = 0.029$ | | |
| Shoulder (flex/ext) | -10.42 | 4.04 | 18.50 | -11.87 | 2.61 | 17.09 | -23.74 | 1.92 | 27.58 |
| correlation (heteroscedasticity) | $p < 0.001$ | $r = -0.53$ | | $p < 0.001$ | $r = -0.39$ | | $p < 0.001$ | $r = -0.39$ | |
| coefficient of determination | $r^2 = 0.28$ | | | $r^2 = 0.15$ | | | $r^2 = 0.16$ | | |
| Shoulder (abd/add) | -20.59 | 9.07 | 38.73 | -15.39 | 12.16 | 39.72 | -69.77 | -7.52 | 54.72 |
| correlation (heteroscedasticity) | $p < 0.001$ | $r = -0.17$ | | $p < 0.001$ | $r = -0.24$ | | $p < 0.001$ | $r = -0.24$ | |
| coefficient of determination | $r^2 = 0.028$ | | | $r^2 = 0.057$ | | | $r^2 = 0.058$ | | |
| Hip (flex/ext) | -41.00 | -0.56 | 39.88 | -56.02 | -5.21 | 45.59 | -29.08 | -4.11 | 20.85 |
| correlation (heteroscedasticity) | $p < 0.001$ | $r = -0.59$ | | $p < 0.001$ | $r = -0.51$ | | $p < 0.001$ | $r = -0.29$ | |
| coefficient of determination | $r^2 = 0.34$ | | | $r^2 = 0.26$ | | | $r^2 = 0.086$ | | |
| Hip (abd/add) | -13.56 | -1.35 | 10.86 | -12.62 | -3.43 | 5.75 | -19.26 | -3.91 | 11.44 |
| correlation (heteroscedasticity) | $p < 0.001$ | $r = -0.40$ | | $p < 0.001$ | $r = -0.31$ | | $p < 0.001$ | $r = -0.60$ | |
| coefficient of determination | $r^2 = 0.16$ | | | $r^2 = 0.096$ | | | $r^2 = 0.36$ | | |
| Knee (flex/ext) | -19.88 | -8.63 | 2.62 | -43.36 | -9.34 | 24.68 | -16.49 | -5.37 | 5.76 |
| correlation (heteroscedasticity) | $p < 0.001$ | $r = -0.50$ | | $p < 0.001$ | $r = -0.16$ | | $p < 0.001$ | $r = -0.16$ | |
| coefficient of determination | $r^2 = 0.25$ | | | $r^2 = 0.025$ | | | $r^2 = 0.025$ | | |
| Ankle (flex/ext) | -36.23 | -22.74 | -9.25 | -46.11 | -17.15 | 11.81 | -29.63 | -17.14 | -4.65 |
| correlation (heteroscedasticity) | $p < .001$ | $r = -.31$ | | $p < .001$ | $r = -.61$ | | $p < .001$ | $r = -.37$ | |
| coefficient of determination | $r^2 = .097$ | | | $r^2 = .37$ | | | $r^2 = .14$ | | |

**Table 6. Limits of agreement and mean bias for all movements with distance.**

| Joint Angle | Distance walk | | | Distance jog | | | Distance wrist shot | | |
|---|---|---|---|---|---|---|---|---|---|
| | Lower limit | Mean bias | Upper limit | Lower limit | Mean bias | Upper limit | Lower limit | Mean bias | Upper limit |
| Elbow (flex/ext) | -35.47 | -21.17 | -6.88 | -28.02 | -12.22 | 3.57 | -41.82 | -20.62 | 0.57 |
| correlation (heteroscedasticity) | $p < 0.001$ | | $r = -0.23$ | $p < 0.001$ | | $r = -0.43$ | $p = 0.044$ | | $r = -0.014$ |
| coefficient of determination | $r^2 = 0.056$ | | | $r^2 = 0.19$ | | | $r^2 = 0.00019$ | | |
| Shoulder (flex/ext) | -3.49 | 7.26 | 18.01 | -8.31 | 3.54 | 15.39 | -26.81 | -1.42 | 23.98 |
| correlation (heteroscedasticity) | $p < 0.001$ | | $r = -0.30$ | $p < 0.001$ | | $r = -0.24$ | $p < 0.001$ | | $r = -0.44$ |
| coefficient of determination | $r^2 = 0.090$ | | | $r^2 = 0.056$ | | | $r^2 = 0.19$ | | |
| Shoulder (abd/add) | -22.63 | 11.10 | 44.82 | -22.24 | 9.80 | 41.85 | -61.27 | 2.88 | 67.03 |
| correlation (heteroscedasticity) | $p < 0.001$ | | $r = -0.44$ | $p < 0.001$ | | $r = -0.31$ | $p < 0.001$ | | $r = 0.31$ |
| coefficient of determination | $r^2 = 0.19$ | | | $r^2 = 0.099$ | | | $r^2 = 0.10$ | | |
| Hip (flex/ext) | -55.40 | -11.18 | 33.05 | -40.31 | -5.24 | 29.82 | -50.33 | -7.42 | 35.49 |
| correlation (heteroscedasticity) | $p < 0.001$ | | $r = -0.69$ | $p < 0.001$ | | $r = -0.70$ | $p < 0.001$ | | $r = -0.56$ |
| coefficient of determination | $r^2 = 0.47$ | | | $r^2 = 0.49$ | | | $r^2 = 0.32$ | | |
| Hip (abd/add) | -12.88 | -1.79 | 9.29 | -12.54 | -1.71 | 9.12 | -16.17 | -1.49 | 13.19 |
| correlation (heteroscedasticity) | $p < 0.001$ | | $r = -0.059$ | $p < 0.001$ | | $r = -0.039$ | $p < 0.001$ | | $r = -0.61$ |
| coefficent of determination | $r^2 = 0.0034$ | | | $r^2 = 0.15$ | | | $r^2 = 0.37$ | | |
| Knee (flex/ext) | -22.64 | -7.72 | 7.19 | -18.96 | -6.64 | 5.69 | -20.74 | -7.53 | 5.67 |
| correlation (heteroscedasticity) | $p = 0.029$ | | $r = -0.14$ | $p < 0.001$ | | $r = -0.29$ | $p < 0.001$ | | $r = -0.14$ |
| coefficient of determination | $r^2 = 0.018$ | | | $r^2 = 0.087$ | | | $r^2 = 0.019$ | | |
| Ankle (flex/ext) | -34.05 | -19.85 | -5.65 | -40.74 | -17.10 | 6.71 | -36.98 | -20.13 | -3.28 |
| correlation (heteroscedasticity) | $p < .001$ | | $r = -.38$ | $p < .001$ | | $r = -.47$ | $p < .001$ | | $r = -.35$ |
| coefficient of determination | $r^2 = .14$ | | | $r^2 = .22$ | | | $r^2 = .12$ | | |

movements while LOA (%) for normalized values was higher with almost all being above 95% with the exception of shoulder abduction/adduction during distance wrist shots at 94.47% and ankle flexion/extension during distance walk at 94.85%. Fig 3 shows the comparison between Bland-Altman plots using raw and normalized joint angles. In addition, it was observed that joint angle wave characteristics were more similar when using normalized rather than the raw joint angles. An example of this is depicted in Fig 4A and 4B.

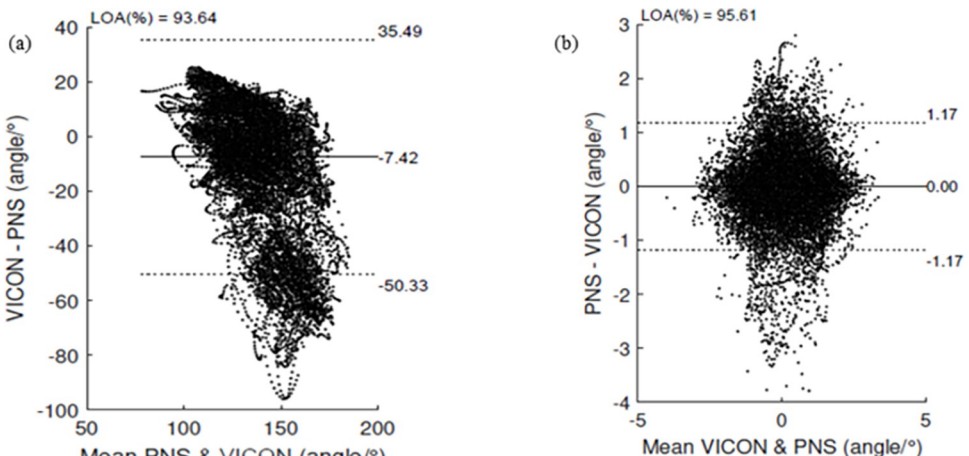

**Fig 3.** Bland-Altman plot of agreement for hip flexion/extension for distance wrist shot using (a) raw joint angles (b) normalized joint angles for all participants. Solid horizontal lines represent the mean difference and the dashed horizontal lines represents the 95% limits of agreement (± 1.96 SD).

**Table 7. LOA (%) comparison between PNS and VICON for raw and normalized values for all movements.**

| Joint Angle | Stationary walk | | Distance walk | | Stationary jog | | Distance jog | | Stationary wrist shot | | Distance wrist shot | |
|---|---|---|---|---|---|---|---|---|---|---|---|---|
| | Raw | Norm | Raw | Norm | Raw | Norm | Raw | Norm | Raw | Norm | Raw | Norm |
| Elbow (flex/ext) | 94.59 | 95.82 | 93.71 | 95.30 | 93.95 | 95.85 | 94.85 | 96.49 | 96.54 | 96.65 | 93.12 | 95.12 |
| Shoulder (flex/ext) | 95.27 | 95.91 | 96.23 | 96.80 | 94.50 | 96.15 | 96.22 | 96.46 | 96.57 | 96.77 | 92.68 | 95.16 |
| Shoulder (abd/add) | 95.41 | 95.36 | 95.26 | 95.48 | 93.70 | 95.85 | 95.59 | 96.70 | 94.81 | 95.48 | 92.61 | 94.47 |
| Hip (flex/ext) | 95.91 | 95.05 | 96.31 | 96.14 | 93.95 | 95.65 | 96.10 | 96.33 | 96.90 | 96.86 | 93.64 | 95.61 |
| Hip (abd/add) | 95.05 | 95.64 | 96.64 | 96.48 | 95.50 | 95.55 | 97.00 | 97.57 | 97.37 | 96.75 | 93.75 | 95.11 |
| Knee (flex/ext) | 95.00 | 95.10 | 94.38 | 95.45 | 94.70 | 95.05 | 94.86 | 96.26 | 97.48 | 96.89 | 94.40 | 95.83 |
| Ankle (flex/ext) | 96.32 | 96.18 | 95.14 | 94.85 | 96.60 | 95.95 | 96.39 | 96.32 | 97.70 | 97.26 | 95.76 | 95.66 |

Table 8 includes the range of movement tested for all joint angles across the different movements.

## Discussion

The purpose of the study was to validate the PNS during full-body human motion capture in comparison to an accepted optoelectronic motion capture systems (i.e., VICON motion analysis system). To establish the validity of the PNS, elbow flexion/extension, shoulder flexion/extension, shoulder abduction/adduction, hip flexion/extension, hip abduction/adduction, knee flexion/extension and ankle flexion/extension angles were calculated from both systems and compared. These joint angles were analysed across movements such as stationary walk, distance walk, stationary jog, distance jog, stationary floorball wrist shot and moving wrist shot. Statistical analyses included RMSE, Pearson's R correlation and Bland-Altman analysis.

Overall, all joint angles across all movements had a RMSE of < 4°, except for ankle flexion/extension during stationary jog and distance jog which had a RMSE of 4.33 and 4.47 respectively and also for shoulder abduction/adduction which had a large RMSE above 11° for both stationary and distance wrist shots. Pearson's R correlation also showed strong relationship for most joint angles except for shoulder flexion/extension during distance walk, hip abduction/adduction during stationary walk and ankle flexion/extension during stationary walk, stationary jog and stationary wrist shot.

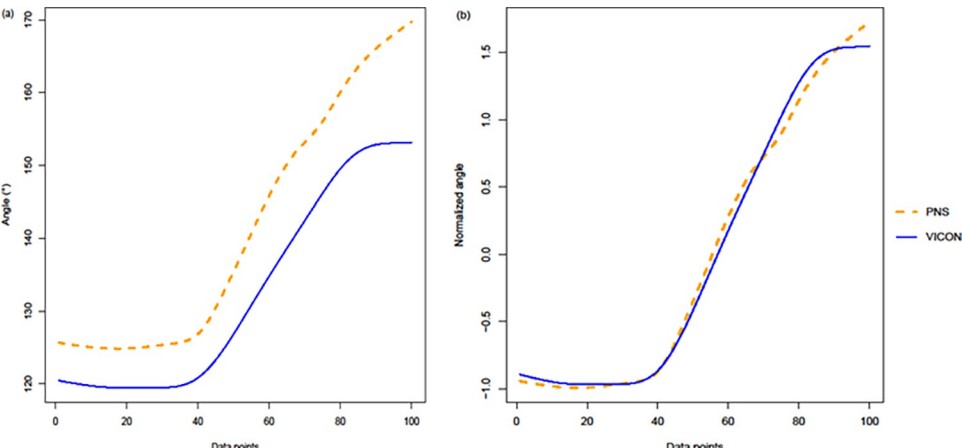

**Fig 4.** Hip flexion/extension angles during a stationary wrist shot using raw joint angle values (a) and Hip flexion/extension angles during a stationary wrist shot using normalized joint angle values (b). Data shown is from one trial from one participant.

**Table 8. Range of joint angles tested for each movement.**

| Joint Angle | Stationary walk | | Stationary jog | | Stationary wrist shot | | Distance walk | | Distance jog | | Distance wrist shot | |
|---|---|---|---|---|---|---|---|---|---|---|---|---|
| | Lower | Upper | Lower | Upper | Lower | Upper | Lower | Upper | Lower | Upper | Lower | Upper |
| Elbow (flex/ext) | 38 | 170 | 39 | 154 | 60 | 165 | 105 | 170 | 47 | 168 | 54 | 158 |
| Shoulder (flex/ext) | 17 | 67 | 20 | 65 | 19 | 93 | 12 | 73 | 22 | 77 | 14 | 101 |
| Shoulder (abd/add) | 33 | 179 | 41 | 179 | 41 | 179 | 106 | 183 | 99 | 181 | 21 | 179 |
| Hip (flex/ext) | 101 | 171 | 86 | 165 | 92 | 172 | 45 | 175 | 94 | 175 | 87 | 175 |
| Hip (abd/add) | 82 | 107 | 83 | 109 | 76 | 133 | 35 | 109 | 82 | 113 | 74 | 126 |
| Knee (flex/ext) | 61 | 172 | 49 | 169 | 108 | 175 | 41 | 177 | 49 | 172 | 64 | 174 |
| Ankle (flex/ext) | 80 | 111 | 70 | 120 | 76 | 127 | 74 | 122 | 73 | 125 | 72 | 124 |

Specifically, values for knee flexion/extension across all movements were acceptable, having an average RMSE of 2.66˚ and a strong relationship with the highest average Pearson's R correlation value of 0.93 among all the other joints angles. There was overall good agreement between the two measurement systems for hip flexion/extension as well with an average Pearson's correlation value of 0.84 across all movements and all movements had a RMSE of $< 4˚$. RMSE for ankle flexion/extension was below 5˚ and Pearson's correlation ranged from moderate to strong across all movements. This indicates that the overall waveform for knee flexion/extension and hip flexion/extension is similar between the two measurement systems. These results were in accordance to results from other studies comparing IMU measurements against VICON motion capture systems. Nüesch, Roos [39] compared the sagittal plane ankle, knee and hip kinematics collected from an inertial sensor system and the VICON motion capture system. Kinematic data was collected during walking and running trials and reported that RMSE was found to be below 5˚ for walking trials and below 8˚ for running trials across the different joint angles. In a study conducted by Bolink, Naisas [40], the use of an IMU was evaluated against the VICON motion capture system during three different activities of daily life: gait, sit-to-stand transfers and block step-up transfers. It was found that the IMU was a valid tool to measure dynamic pelvic angles with a strong Pearson's correlation ranging between 0.85 and 0.94 and RMSE values between 2.7˚ to 4.5˚. Seel, Raisch [41] compared knee flexion/extension measurements between IMU and VICON motion capture systems during gait trials and found RMSE ranging from 1.62–3.3˚. Even though Nüesch, Roos [39] reported a higher measurement error as a result of the increase in the speed of movement from walking to running, this difference was not evident in the current study possibly because walking and running speeds were self-selected.

In general, results for upper body joint angles were not as good as compared to lower body joint angles. This is in line with systematic reviews [42, 43] which reported that the large variability observed in the upper body measurements could be due to how functional upper body tasks usually involves complex movements within two-three axes. Even with this, the magnitude of error for elbow flexion/extension and shoulder flexion/extension were still comparable to those reported in previous studies. Pearson's R correlation (0.84 to 0.96) and RMSE ($< 4˚$) for elbow flexion/extension measurements found in this study was similar to previous studies done [42, 44, 45]. Shoulder (flex/ext) also had a moderate to strong Pearson's R correlation and an RMSE of $< 3˚$ across all movements which is within the range of error found in other studies [18, 42].

Results for shoulder abduction/adduction differed most from previous studies. Pearson' R correlation was found to range between moderate to strong (0.58 to 0.87) with lower values during stationary and distance wrist shots. RMSE was also higher than what was reported for other joints, ranging from 5.36˚ to 15.15˚. Similar to Pearson's R correlation, RMSE values

were also higher during stationary and distance wrist shots. Most studies have suggested that any RMSE up to 5˚ can be regarded as reasonable and are precise enough for a system to be considered as a valid method for motion analysis in clinical settings. However, when errors are above 5˚, it should raise concerns as it is considered large enough to mislead clinical interpretation [12, 40, 43, 46, 47]. Therefore the use of raw shoulder abduction/adduction joint angles should be treated with caution, with the degree of acceptable limits directly dependent on the intended application [48].

Bland-Altman analysis showed that the mean systematic bias was below 10˚ for most joint angles with the exception of the elbow flexion/extension for all movements, shoulder abduction/adduction for distance walk, hip flexion/extension during distance walk and ankle flexion/extension for all movements (Tables 5 and 6). Bland-Altman plots for all raw and normalized joint angles can be found in Fig 3. Based on the recommendation of previous studies, a 10˚ lower or upper LOA was deemed acceptable for occupational biomechanics applications [18, 49, 50]. Therefore, joint angles obtained from the PNS seems satisfactory except for elbow flexion/extension for all movements, ankle flexion/extension for all movements, shoulder abduction/adduction and hip flexion/extension during distance walk for which data should be handled with discretion especially during dynamic movements when absolute measurement of joint angles is required.

Results showed significant correlation between the mean joint angle value and the difference between the two measurements system for almost all the joints and across movements. Therefore, there is a possibility of heteroscedasticity [38] where the amount of error from the PNS might increase as the measured joint angle value increases. Significant correlation was found even with a small coefficient of determination ($r^2$) between the mean joint angles and difference between the two measurements. The observed significance despite a very small $r^2$ is most probably due to the large sample size (above 19000 data points). However, validation is still confirmed for the range of movement that was investigated in this study (see Table 8), but caution should be applied when analyzing larger joint angles (when higher values are possible).

When using normalized and raw joint angles for comparison between the two systems, LOA (%) results were consistently higher for normalized values (range from 94.47% to 97.57%) as compared to raw joint angles (range from 92.49% to 97.52%). In addition, it was observed that joint angle wave characteristics were more similar when using normalized rather than the raw joint angles. The normalization process of the joint angles managed to preserve the waveform characteristics (e.g., peak joint moment) and reduce the variance between the two systems. As such, it is suggested that when using the PNS, the analysis of normalized joint angles is preferred instead of the raw joint angles as the PNS is better at detecting changes in movement patterns rather than to approximate the absolute joint angles.

## Limitations

A limitation of the current study is the small collection time frame for each trial. It has been reported in previous studies that IMUs accuracy differed accordingly to the type of tasks being performed and is affected by the duration, complexity and speed of the task performed [18, 39, 42, 51–54]. Accuracy of IMUs is also affected by time of the recorded task. Its accuracy decreases over time as a result of drift which continually increases and the amount of error is compounded over time [54, 55]. Drift occurs due to the magnetic perturbations in the environment around the sensors and also the ability of the sensors to locate and track the same initial frames [56]. The amount of drift is dependent on the system and the direction of movement, with accuracy decreasing more for movements that occur within 3-axes over a

sustained period of time [56]. Data included in the current study was collected within 20–30 seconds before the next calibration. Therefore, future studies could include longer trials to detect how much the PNS drifts over time and its resilience against magnetic perturbation in order to provide more information on its accuracy and if any correction is needed in the algorithms.

Previous studies have found that the accuracy of IMUs is negatively influenced by speed [57]. Larger RMSE was reported by Nüesch, Roos [39] during running as compared to walking on a treadmill, with errors of $< 8˚$ and $< 5˚$ respectively. Similarly, Cooper G., Sheret I. [51] found a RMSE of $< 4˚$ during running and $< 1˚$ of the knee during walking. However, the effect of speed on the accuracy of the PNS was not as evident as in other studies. Comparisons between walking and jogging tasks did not elicit a significant change in the results. One possible reason could be that the walking and jogging speeds were self-selected and not controlled, therefore the walking and jogging speeds might not have been significantly different enough to result in a difference. This result was similar to the study conducted by Sers, Forrester [12] where no obvious difference was detected with a change in movement speeds as speeds were also self-selected. Therefore, future studies could also analyse the accuracy of the PNS with movements at different controlled speeds. In the current study transverse plane data was not analysed, future studies could look into including comparisons between the two systems within the transverse plane.

Finally, in this study the impact on accuracy from the suit on the soft tissue artifact from the markers was not accounted for which could be a potential source of uncertainty. Therefore, future research could use rigid marker clusters that are fixed to the PNS to minimize the differences between the two systems associated with soft tissue artifact.

## Conclusion and recommendations

The validity of the PNS was tested against the VICON motion analysis system by comparing joint angle measurements obtained by the two systems during a series of full-body human movements. In general, the PNS performed well against the VICON motion analysis system and most joint comparisons were similar to that of what was reported in previous studies, with the exception of elbow flexion/extension, shoulder abduction/adduction during the stationary and distance wrist shots, hip flexion/extension during distance walk and ankle flexion/ extension.

With reference to Bland-Altman analysis, most of the joint angles were found to have been overestimated by the PNS with the exception of the shoulder flexion/extension and shoulder abduction/adduction. Mean bias for most joint angles were within acceptable recommendations from previous studies [18, 49, 50]. LOA range when using raw joint angles was larger than what is recommended for biomechanics applications [18, 49, 50]. Thus, caution should be applied when using raw joint angle measurements obtained from the PNS for analysis. In addition, validation of the PNS is confirmed for the range of movements that were investigated in this study and caution should also be applied when analyzing any joint angles beyond this range. When using normalized joint angles for comparisons, LOA (%) values were consistently better as compared to when raw joint angles were used. In conclusion, results suggests that the PNS could be more suited to analyse movement patterns through the use of normalized rather than raw joint angles and that researchers could consider the use of the PNS when doing so.

## Supporting information

**S1 Appendix. Bland-Altman plots.**
(PDF)

## Author Contributions

**Conceptualization:** Corliss Zhi Yi Choo, Jia Yi Chow.

**Data curation:** Corliss Zhi Yi Choo.

**Formal analysis:** Corliss Zhi Yi Choo, John Komar.

**Investigation:** Corliss Zhi Yi Choo.

**Methodology:** Corliss Zhi Yi Choo, Jia Yi Chow.

**Project administration:** Corliss Zhi Yi Choo, Jia Yi Chow.

**Resources:** Corliss Zhi Yi Choo.

**Software:** Corliss Zhi Yi Choo, John Komar.

**Supervision:** Jia Yi Chow, John Komar.

**Validation:** Corliss Zhi Yi Choo.

**Writing – original draft:** Corliss Zhi Yi Choo.

**Writing – review & editing:** Corliss Zhi Yi Choo, Jia Yi Chow, John Komar.

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
