## [Decision Letter · Decision Letter 0]

26 Jan 2021

PONE-D-20-33330

Validation of the Perception Neuron system for full-body motion capture

PLOS ONE

Dear Dr. Choo,

Thank you for submitting your manuscript to PLOS ONE. After careful consideration, we feel that it has merit but does not fully meet PLOS ONE’s publication criteria as it currently stands. Therefore, we invite you to submit a revised version of the manuscript that addresses the points raised during the review process.

We look forward to receiving your revised manuscript.

Kind regards,

Tumay Tunur, Ph.D.

Academic Editor

PLOS ONE

Reviewers' comments:

Reviewer's Responses to Questions

**Comments to the Author**

1. Is the manuscript technically sound, and do the data support the conclusions?

Reviewer #1: Partly

Reviewer #2: Partly

2. Has the statistical analysis been performed appropriately and rigorously? 

Reviewer #1: Yes

Reviewer #2: Yes

3. Have the authors made all data underlying the findings in their manuscript fully available?

Reviewer #1: No

Reviewer #2: Yes

4. Is the manuscript presented in an intelligible fashion and written in standard English?

Reviewer #1: Yes

Reviewer #2: Yes

5. Review Comments to the Author

Reviewer #1: This is a paper where the authors have tried to validate the PNS IMU-based system kinematic output against that of the VICON optical capture system. The need for their work is that validation of the PNS system is currently limited. The significance of their work is that, if the PNS system kinematic output is validated, then one could conduct motion analysis studies by using a less expensive system which is easier to transport, it does not require a lot of time to set up and can be used outside of a laboratory environment in a real-world setting. However, besides the cost-related and portability aspects of the authors’ arguments in favor of the PNS system, there are multiple examples of optical capture systems used in a real-world setting, either for studying sports performance or occupational factors. Furthermore, the claim of the authors that set up and calibration is very lengthy for optical capture systems, is not in agreement with our experience, assuming knowledge of optical capture principles. Our volume calibration procedure takes approximately 1 minute consistently, and within the calibration volume one can perform any activity. The marker-model calibration, which is specific to the computational model used to analyze motion, is done once, also done in less than 1 minute (6 seconds to be exact) rather than re-calibrating every time a new activity is to be performed, which seems to be the case with the PNS. However, the authors do claim to be calibrating both systems before each activity (page 8, line 167), which does not make sense with respect to the VICON system.

There are some major concerns before this manuscript is considered for publication. Some of them are:

1) It seems that the output from each system was filtered in the exact same way (page 10, lines 195-197). How did the authors determine that the frequency content to be filtered from both systems for each activity was identical and, therefore, presumably they used the same filtering process and parameters? Data need to be provided towards this end. If this comment is not correct, then how each signal output from each device for each activity was filtered and why? This information is directly related to the quality of the data output being compared.

2) The authors have used the Plug-In-Gait marker set to compute the kinematics for the activities they studied (page 7, lines 161-162). The Plug-In-Gait marker set, however, is tied to several assumptions tied to the Plug-In-Gait computational model. It appears, however, that the authors did not use the Plug-In-Gait model to compute the kinematics of interest (page 10, lines 187-191). Why did the authors use this marker set model? The marker model itself does not ensure 3D segment definition by 3 marker non-collinearity per segment. Therefore, what assumptions did the authors use for the purposes of 3D segment definition (especially for the femur and tibia/fibula) and corresponding 3D motion computation? This information is also directly related to the quality of the data output being compared.

3) Observing Figure 2, it appears that pelvis is defined by the 2 ASIS markers and the 2 PSIS markers. These markers appear to have been placed on the shorts/trunks of the subject, presumably directly over the specific anatomical landmarks (it is not clear, but it appears that a mid-thigh marker is also placed on the subject’s trunks). How did the authors made certain that all these markers on the trunks remained over the anatomical landmarks of interest throughout the duration of all activities after each calibration? This information is also directly related to the quality of the data output being compared.

4) Observing Figure 2, it appears that the mid-thoracic marker (typically placed on the 10th thoracic spinous process) is placed neither on the 10th thoracic spinous process, nor in line with the thoracic spinous processes. This would create a trunk orientation offset. How has such an offset been accounted for during the kinematic computation of the shoulder motion, which, presumably is measured relative to the trunk?

5) Transverse plane data are very important, especially when considering injury prevention (in sports and occupational biomechanics) and in the clinical decision-making process. Given the ambitions of the authors for the PNS outlined in the introduction, the authors need to provide transverse data output comparisons between the 2 systems.

6) Observing Figures 1 and 2, it appears that PNS can provide ankle-related kinematics. Given the importance of ankle kinematics, especially for all lower extremity activities (in walking gait approximately 80% of the ambulatory power generation is related to the ankle), and given the desire of the authors to involve functional activities in this study, ankle kinematic output comparisons need to be made, even if only for the sagittal plane and consistently with the study of Nuesch and Ross.

7) From Tables 3 and 4 it appears that for the same activity the r and RMSE increased and decreased, respectively, for most activities as a function of increasing speed. The authors suggest that this may be related to subjects performing these activities at self-selected speeds. Although performing an activity at a self-selected speed, indeed, has been found to decrease the within trial or within condition/speed variability, the mere increase in velocity results in higher kinematic variability in VICON even from a motion artifact point of view. Therefore, the authors’ argument of self-selected speed is ambiguous. On the contrary, this suggests that either PNS’s performance for activities that are static or quasi static is substandard or the earlier arguments regarding the filtering process need to be given serious consideration.

8) What is the normalization process implemented in the Bland altman plots?

Consequently, for all these major reasons I think this manuscript is not appropriate for publication in the current time and in the current form.

Reviewer #2: The goal of this study, the evaluation of an IMU system for the measurement of joint angles, is worthy and timely as both hardware and software/algorithms for the detection of human movement have improved. Additionally, this study looks at 3d motion in contrast to earlier work which has evaluated the system in 2d only. Likewise, multiple ranges of motion and activities were evaluated by this work.

The work shows some thoroughness in scope, although that is not always reflected in the presentation. In particular, certain key details are inconsistent between text and tables, and not all the information that is collected is presented. This is easily remedied, I think, as I note below. There are also some minor grammatical issues or issues of clarification that noted.

Introduction

Lines 52-53

Change to “reliable and suitable methods to measure complex movements and are often considered as the gold standard in motion capture, providing an estimated accuracy (RMS error) of less than 1.00° and 1.50°” For RMS errors, smaller is better, so “less than” seems more appropriate than “up to”.

Line 60

Oxford comma after “calibration techniques”

Line 64

Due to “their validity” since we are discussing multiple systems.

Lines 72-77

In modern optical systems having 8+ cameras, occlusion is not generally a problem. It used to be a problem in the past. Where it can still arise is if the subject is wearing loose clothing. Please clarify this or give a specific example.

Lines 89-90

You mention increased accuracy of IMUs. But what is this compared to? Do you mean earlier IMU systems or optical systems? Some Mocap systems can do 1000 fps (see for example: https://www.xcitex.com/procapture-high-speed-cameras.php

https://arxiv.org/pdf/1908.11505.pdf

https://www.norpix.com/blog/high-speed-system-captures-at-1000-fps-x-1080p-from-multiple-synchronized-cameras/

and my own very old Motion Analysis Corp system can do 500 fps.

Line 124

You use the word “biasness.” It is a word, but I suspect that the word “bias” is in more common use. I leave this to your judgment.

Line 133

Oxford comma after “jogging”

Lines 137-138

Be consistent and use either PN system(s) or PNS throughout the paper.

Participants

Line 149

When you start a sentence with numbers, please spell them out unless you have style guidelines from PLOS that say differently.

Line 152

“musculoskeletal injury, illness, or disease” – note the change in plurality and Oxford comma.

Instrumentation

Line 159

Vicon is a very widely used system, but it’s not clear to me that it is used more universally than Qualisys, MAC, or others. Perhaps you could just indicate that it is widely used rather than “the most widely used” unless you are prepared to cite hard data.

Line 161

When you start a sentence with numbers, please spell them out.

Line 163 – This is just a curiosity and does not need to be addressed in the paper – is there any work showing how IMU suits affect soft tissue movement?

Line 164 – Also a curiosity – is there any work looking at how trailing cables interfere with movement? This should not affect your results one way or another, but it could be important using such a system clinically.

Data Analysis

Line 190

Mathworks is in Natick, MA, not Natrick.

Line 202

Please provide a reference. Are these Cohen’s values?

Line 206

Please indicate that X and Y are time series in the above equation and in the text by using notation such as X(t) and Y(t). Indicate summation over time.

Results

Line 227

Distance wrist shot shoulder angles also did not have a strong correlation according to your table.

Line 238

Your Figure 3 only includes data for hip flexion/extension for a wrist shot.

Discussion

Here there are no line numbers. You do not mention hip ab/adduction during walking.

Please explain in more detail why the hip ab/adduction during walking is not so accurate. Does this have to do with the positioning of the IMUs, lower spinal flexion that is not captured by this setup, or some other reason?

Conclusion and recommendations

(use and not &)

Note (end of first paragraph) that hip ab/adduction during walking also does not match Vicon.

Plotting

The plotting comparing joint angles is only for hip flexion in a single activity for a single trial for a single subject. Although there are supporting Bland-Altman plots, I suspect that the biomechanics community would like to see some sort of ensemble comparison of joint angles for each activity. PLOS ONE has no limitation on the number of figures. However, I would consider how to standardize the data so that information for the same activity from multiple subjects could be placed on the same figure. Likewise, it is possible to do multipanel figures – for example – showing multiple ranges of motion across multiple subjects in a single figure. The point of this is to present a convincing case with supporting materials that your results are as advertised.

6. PLOS authors have the option to publish the peer review history of their article (what does this mean?). If published, this will include your full peer review and any attached files.

Reviewer #1: No

Reviewer #2: No

---

## [Author Response · Author response to Decision Letter 0]

8 Mar 2021

Thank you for the positive review and constructive comments. Below, we highlight the revisions made to enhance the manuscript. The information stated below has also been attached as a file named: "Response to reviewers".

Pg 3, row 52.

Replace method with methods. 

This has been revised. See pg 3, row 52.

Pg 3, row 53.

Replace up to < with less than. 

This has been revised. See pg 3, row 53 to 54.

Pg 3, row 60.

Insert comma. 

This has been revised. See pg 3, row 60.

Pg 3, row 64.

Replace its with their. 

This has been revised. See pg 3, row 64. 

Pg 4, row 72-77

Comment: Although true, most systems have enough cameras that this is not a problem… For most lab collections obstruction is not an issue. Is there a specific application you have in mind where it is? 

The purpose of the study was to investigate and potentially provide evidence for researchers that the PNS could be a viable tool to collect motion capture data in a more representative environment outside of a lab setting. This is especially so when it is not feasible to set up an optical based system. For example in situations where there is a need to analyse movement over a larger area, if an optical based system is used, a greater number of cameras would be required to cover this increase in volume. In this case, the PNS could be used to overcome the limitations of the lab space and the number of cameras needed. 

Pg 4, row 89-90.

Comment: Is this compared to earlier IMUs? What’s the accuracy? You cited the accuracy of the optical mocap systems earlier. 

Yes, this is in comparison with earlier IMUs. As the accuracy of the sensors are highly dependent on the sensor specifications, software algorithms, movement type and joint angle analysed there is a large range for this value. Therefore, no exact value was indicated in this section. This was highlighted on pg 5, row 106 to 109.

Pg 6, row 124.

Comment: Uncommon word, bias is probably preferable. This has been revised. See pg 6, row 124. 

Pg 6, row 133.

Insert comma. 

This has been revised. See pg 6, row 133.

Pg 6, row 138. 

Comment: Be consistent with the use of PNS. 

This has been revised. Has been changed to PNS for these instances: 

Pg 6, row 137; pg 8, row 175: pg 9, row 180; pg 11, row 206.

Pg 7, row 149.

Comment: Spell out the number. 

This has been revised. See pg 7, row 149. 

Pg 7, row 152.

Change illnesses to illness. 

This has been revised. See pg 7, row 152.

Pg 7, row 152.

Change diseases to disease.

This has been revised. See pg 7, row 153.

Pg 7, row 159.

Change from the most widely used to a widely used system? 

This has been revised. See pg 7 row 159.

Pg 7, row 160. 

Comment: Was the statement “The VICON optoelectronic motion analysis system was chosen as it is a widely used system and is considered as the current laboratory gold-standard with a high accuracy during dynamic trial.” the result of Topley & Richards (2020)? 

Topley & Richards (2020) did not specifically mention that the VICON was a widely used system or that it was considered as the current lab gold standard. It did indicate the high accuracy of the VICON and other camera systems, however testing was conducted using the surface of a rigid aluminum arm that rotates. Therefore, this reference was not added at the end of the sentence since it did not support the statement.

Pg 7, row 161.

Spell out the number in the sentence. 

This has been revised. See pg 7, row 161.

Pg 7, row 163.

Comment: Does the suit increases or decrease the soft tissue artifact from the markers? Is this a source of uncertainty? 

The amount of error resulting from this was not accounted for in the current study. Therefore, It has been included as a limitation, see pg 24, row 374 to 377.

Pg 8, row 164. 

Comment: Does the USB connection affect peoples’ movement as old EMG system wires used to? The wires connecting between each sensor uses a coiled design, so it allows participants to stretch and move through the range of movement freely when necessary but at the same time is still neatly kept which prevents any excessive wiring from dangling around to potentially impede movement. The system connects through a single wire to the laptop and this wire can be placed to face different directions and be shifted out of the way to accommodate various movements. 

Pg 10, row 190. 

Change Natrick to Natick. 

This has been revised. See pg 10, row 190.

Pg 10, 202. 

Comment: Reference? Is this Cohen? 

The citation has been added. See pg 10, row 203.

Pg 11 row, 206. 

Comment: Change X to X(t) and Y to Y(t). Indicate that these are time dependent and that you are summing over time. 

This has been revised. See pg 11, row 206 to 207.

Pg 12, row 227.

Comment: Distance wrist shot shoulder also not a strong correlation. 

This has been added. See pg 12, row 229 to 230. 

Pg 12, row 238. 

Comment: This is only for flexion/extension for the wrist shot. 

This has been revised to state that the figure is only indicative for hip (flexion/extension) during distance wrist shot. See pg 12, row 241 to 243. 

Pg 13, row 244. 

Comment: How were data normalized? 

Data was normalized using z-score transformation. Details have been added to pg 11, row 218 to 221.

Pg 22. 

Comment: What about hip ab/ad?

The mean systematic bias for hip flexion/extension during distance walk was found to be -11.18°. This has been included. Hip abduction/adduction was -1.35 for stationary walk and -1.79° for distance walk. See pg 22, row 323 to 324, pg 323 row 328 to 329, and pg 24 row 383 to 384.

Pg 24.

Change & to and. 

This has been revised. See pg 24, row 377 and pg 24, row 378.

---

## [Decision Letter · Decision Letter 1]

26 Apr 2021

PONE-D-20-33330R1

Validation of the Perception Neuron system for full-body motion capture

PLOS ONE

Dear Dr. Choo,

Thank you for submitting your manuscript to PLOS ONE. After careful consideration, we feel that it has merit but does not fully meet PLOS ONE’s publication criteria as it currently stands. Therefore, we invite you to submit a revised version of the manuscript that addresses the points raised during the review process.

ACADEMIC EDITOR: As far as we see, you only responded to the reviewer 2's comments in the previous round, so the reviewer 1's comments remain un-responded. In this round, please respond to both reviewers' comments separately. Also, I would like to suggest to copy-paste the reviewers' comments as they are in the response letter, instead of summarizing their comments, such that we can see which comments you are responding.

We look forward to receiving your revised manuscript.

Kind regards,

Kei Masani

Academic Editor

PLOS ONE

Reviewers' comments:

Reviewer's Responses to Questions

**Comments to the Author**

1. If the authors have adequately addressed your comments raised in a previous round of review and you feel that this manuscript is now acceptable for publication, you may indicate that here to bypass the “Comments to the Author” section, enter your conflict of interest statement in the “Confidential to Editor” section, and submit your "Accept" recommendation.

Reviewer #1: (No Response)

Reviewer #2: All comments have been addressed

2. Is the manuscript technically sound, and do the data support the conclusions?

Reviewer #1: No

Reviewer #2: Partly

3. Has the statistical analysis been performed appropriately and rigorously? 

Reviewer #1: Yes

Reviewer #2: Yes

4. Have the authors made all data underlying the findings in their manuscript fully available?

Reviewer #1: No

Reviewer #2: No

5. Is the manuscript presented in an intelligible fashion and written in standard English?

Reviewer #1: Yes

Reviewer #2: Yes

6. Review Comments to the Author

Reviewer #1: Although this is a re submission, the authors have not addressed any of my comments from the original review. I am re-inserting them for completion.

This is a paper where the authors have tried to validate the PNS IMU-based system kinematic outpout against that of the VICON optical capture system. The need for their work is that validation of the PNS system is currently limited. The significance of their work is that, if the PNS system kinematic output is validated, then one could conduct motion analysis studies by using a less expensive system which is easier to transport, it does not require a lot of time to set up and can be used outside of a laboratory environment in a real-world setting. However, besides the cost-related and portability aspects of the authors’ arguments in favor of the PNS system, there are multiple examples of optical capture systems used in a real-world setting, either for studying sports performance or occupational factors. Furthermore, the claim of the authors that set up and calibration is very lengthy for optical capture systems, is not in agreement with our experience, assuming knowledge of optical capture principles. Our volume calibration procedure takes approximately 1 minute consistently, and within the calibration volume one can perform any activity. The marker-model calibration, which is specific to the compuatational model used to analyze motion, is done once, also done in less than 1 minute (6 seconds to be exact) rather than re-calibrating every time a new activity is to be performed, which seems to be the caase with the PNS. However, the authors do claim to be calibrating both systems before each activity (page 8, line 167), which does not make sense with respect to the VICON system.

There are some major concerns before this manuscript is considered for publication. Some of them are:

1) It seems that the output from eah system was filtered in the exact same way (page 10, lines 195-197). How did the authors determine that the frequency content to be filtered from both systems for each activity was identical and, therefore, presumably they used the same filtering process and parameters? Data need to be provided towards this end. If this comment is not correct, then how each signal output from each device for each activity was filtered and why? This information is directly related to the quality of the data output being compared.

2) The authors have used the Plug-In-Gait marker set to compute the kinematics for the activities they studied (page 7, lines 161-162). The Plug-In-Gait marker set, however, is tied to several assumptions tied to the Plug-In-Gait computational model. It appears, however, that the authors did not use the Plug-In-Gait model to compute the kinematics of interest (page 10, lines 187-191). Why did the authors use this marker set model? The marker model itself does not ensure 3D segment definition by 3 marker non-colinearity per segment. Therefore, what assumptions did the authors use for the purposes of 3D segment definition (especially for the femur and tibia/fibula) and corresponding 3D motion computation? This information is also directly related to the quality of the data output being compared.

3) Observing Figure 2, it appears that pelvis is defined by the 2 ASIS markers and the 2 PSIS markers. These markers appear to have been placed on the shorts/trunks of the subject, presumably directly over the specific anatomical landmarks (it is not clear, but it appears that a mid-thigh marker is also placed on the subject’s trunks). How did the authors made certain that all these markers on the trunks remained over the anatomical landmarks of interest throughout the duration of all activities after each calibration? This information is also directly related to the quality of the data output being compared.

4) Observing Figure 2, it appears that the mid-thoracic marker (typically placed on the 10th thoracic spinous process) is placed neither on the 10th thoracic spinous process, nor in line with the thoracic spinous processes. This would create a trunk orientation offset. How has such an offset been accounted for during the kinematic computation of the shoulder motion, which, presumably is measured relative to the trunk?

5) Transverse plane data are very important, especially when considering injury prevention (in sports and occupational biomechanics) and in the clinical decision-making process. Given the ambitions of the authors for the PNS outlined in the introduction, the authors need to provide transverse data output comparisons between the 2 systems.

6) Observing Figures 1 and 2, it appears that PNS can provide ankle-related kinematics. Given the importance of ankle kinematics, especially for all lower extremity activities (in walking gait approximately 80% of the ambulatory power generation is related to the ankle), and given the desire of the authors to involve functional activities in this study, ankle kinematic output comparisons need to be made, even if only for the sagital plane and consistently with the study of Nuesch and Ross.

7) From Tables 3 and 4 it appears that for the same activity the r and RMSE increased and decreased, respectively, for most activities as a function of increasing speed. The authors suggest that this may be related to subjects performing these activities at self-selected speeds. Although performing an activity at a self-selected speed, indeed, has been found to decrease the within trial or within condition/speed variability, the mere increase in velocity results in higher kinematic variability in VICON even from a motion artifact point of view. Therefore, the authors’ argument of self-selected speed is ambiguous. On the contrary, this suggests that either PNS’s performance for activities that are static or quasi static is substandard or the earlier arguments regarding the filtering process need to be given serious consideration.

8) What is the normalization process implemented in the Bland altman plots?

Consequently, for all these major reasons I think this manuscript is not appropriate for publication in the current time and in the current form.

Reviewer #2: Thank you for taking the time to make the recommended changes. I have only a couple more points of clarification listed below. I want to reiterate that as motion capture systems are becoming more mobile, there is a need to evaluate them to extend their applicability and accessibility, and so I feel that your work is timely.

I originally had a number of comments on the rewrite for lines 68-77. In particular these were regarding the ease of setup and the occlusion of markers. I suggest that your comments below address my concerns and that this wording, or something similar should be in the actual paper not just in the response should the editors permit the increase in word count.

"The purpose of the study was to investigate and potentially provide evidence for researchers that the PNS could be a viable tool to collect motion capture data in a more representative environment outside of a lab setting."

"This is especially so when it is not feasible to set up an optical based system. For example in situations where there is a need to analyse movement over a larger area, if an optical based system is used, a greater number of cameras would be required to cover this increase in volume. In this case, the PNS could be used to overcome the limitations of the lab space and the number of cameras needed."

Regarding the instrumentation section – I did not see a frame rate for the IMU system, did I miss it?

Line 177 – the correct number of markers is 33, right?

Data analysis - Since the specific algorithm you used is not completely clear, nor is the method used

to synch the data, I would suggest publishing your code or else describing what was done here in more detail. It’s probably easier to just post the MATLAB code, I would think.

Data analysis - The Sovitzy-Golay filter may not be as well known in the biomechanics community as you think. My recollection is that it is a moving average method involving convolution of low order polynomials. I would describe this more clearly, especially since there are questions about your filter frequency which I don't think specifically applies with this type of smoothing. Since different polynomials may be fit to different portions of the data, it seems that there is non-uniform smoothing which can match the local roughness of the data better and that the “frequency” is polynomical dependent. This is discussed in the white-paper found here:

https://www.hpl.hp.com/techreports/2010/HPL-2010-109.pdf

Alternatively, I would cite a source describing these filters and their response, presuming that the other reviewer is content with this sort of explanation.

Table 3: Generally, correlations are done between series of data. In this case, I would expect a correlation calculated between VICON and IMU data per joint per activity and for each subject. The cited SD is then the SD of correlation coefficients across all subjects, right? Each correlation test has a p-value. What is the p-value shown here if multiple correlation tests were done to determine the mean correlation andits standard deviation? Please clarify.

Finally – I did not find your data in the NIE Data Repository – please post a URL!

7. PLOS authors have the option to publish the peer review history of their article (what does this mean?). If published, this will include your full peer review and any attached files.

Reviewer #1: No

Reviewer #2: No

---

## [Author Response · Author response to Decision Letter 1]

20 Oct 2021

The complete response to reviewers has been attached in the document titled: Response to Reviewers.

---

## [Decision Letter · Decision Letter 2]

14 Dec 2021

PONE-D-20-33330R2Validation of the Perception Neuron system for full-body motion capturePLOS ONE

Dear Dr. Choo,

Thank you for submitting your manuscript to PLOS ONE. After careful consideration, we feel that it has merit but does not fully meet PLOS ONE’s publication criteria as it currently stands. Therefore, we invite you to submit a revised version of the manuscript that addresses the points raised during the review process. I would like to ask to make two minor changes.Please make the minor change suggested by the reviewer 2.Also, please add the description about the compression garments in the manuscript, which you made in the response to the reviewer 1 with the citation of Mills's paper. Please submit your revised manuscript by Jan 28 2022 11:59PM. If you will need more time than this to complete your revisions, please reply to this message or contact the journal office at plosone@plos.org. Please include the following items when submitting your revised manuscript:A rebuttal letter that responds to each point raised by the academic editor and reviewer(s). You should upload this letter as a separate file labeled 'Response to Reviewers'.A marked-up copy of your manuscript that highlights changes made to the original version. You should upload this as a separate file labeled 'Revised Manuscript with Track Changes'.An unmarked version of your revised paper without tracked changes. You should upload this as a separate file labeled 'Manuscript'.If applicable, we recommend that you deposit your laboratory protocols in protocols.io to enhance the reproducibility of your results. Protocols.io assigns your protocol its own identifier (DOI) so that it can be cited independently in the future. For instructions see: https://journals.plos.org/plosone/s/submission-guidelines#loc-laboratory-protocols. Additionally, PLOS ONE offers an option for publishing peer-reviewed Lab Protocol articles, which describe protocols hosted on protocols.io. Read more information on sharing protocols at https://plos.org/protocols?utm_medium=editorial-email&utm_source=authorletters&utm_campaign=protocols.

We look forward to receiving your revised manuscript.

Kind regards,

Kei Masani

Academic Editor

PLOS ONE

Journal Requirements:

Reviewers' comments:

Reviewer's Responses to Questions

**Comments to the Author**

1. If the authors have adequately addressed your comments raised in a previous round of review and you feel that this manuscript is now acceptable for publication, you may indicate that here to bypass the “Comments to the Author” section, enter your conflict of interest statement in the “Confidential to Editor” section, and submit your "Accept" recommendation.

Reviewer #2: All comments have been addressed

2. Is the manuscript technically sound, and do the data support the conclusions?

Reviewer #2: Yes

3. Has the statistical analysis been performed appropriately and rigorously? 

Reviewer #2: Yes

4. Have the authors made all data underlying the findings in their manuscript fully available?

Reviewer #2: No

5. Is the manuscript presented in an intelligible fashion and written in standard English?

Reviewer #2: (No Response)

6. Review Comments to the Author

Reviewer #2: Thank you for the updates. However, your DOI (https://doi.org/10.25340/R4/T6W5RX) is returning a "DOI not found" from doi.org. Please check this and ensure that it is posted correctly in your final draft so that readers of the paper may access this information. This is my sole reason for selecting "Minor Revision" rather than "Accept" at this point.

7. PLOS authors have the option to publish the peer review history of their article (what does this mean?). If published, this will include your full peer review and any attached files.

Reviewer #2: **Yes: **Timothy A Niiler

---

## [Author Response · Author response to Decision Letter 2]

29 Dec 2021

Thank you for the points raised. Below, we highlight the revisions that have been made. 

Reviewer 2 comments: "Thank you for the updates. However, your DOI (https://doi.org/10.25340/R4/T6W5RX) is returning a "DOI not found" from doi.org. Please check this and ensure that it is posted correctly in your final draft so that readers of the paper may access this information. This is my sole reason for selecting "Minor Revision" rather than "Accept" at this point."

Response: Apologies for that. The data can be accessed through this updated link: https://doi.org/10.25340/R4/IZKYZR

However, the link would only be active when the paper is published. 

This has been updated in the submission steps as well, under the section ‘Describe where the data may be found in full sentences. If you are copying our sample text, replace any instances of XXX with the appropriate details.’. 

Academic Reviewer comments: "Please add the description about the compression garments in the manuscript, which you made in the response to the reviewer 1 with the citation of Mills's paper."

Response: The description about the compression garments have been added into the Manuscript on page 9, line 194-198.

---

## [Editor Report · Decision Letter 3]

5 Jan 2022

Validation of the Perception Neuron system for full-body motion capture

PONE-D-20-33330R3

Dear Dr. Choo,

We’re pleased to inform you that your manuscript has been judged scientifically suitable for publication and will be formally accepted for publication once it meets all outstanding technical requirements.

Kind regards,

Kei Masani

Academic Editor

PLOS ONE
---

## [Editor Report · Acceptance letter]

11 Jan 2022

PONE-D-20-33330R3 

Validation of the Perception Neuron system for full-body motion capture 

Dear Dr. Choo:

I'm pleased to inform you that your manuscript has been deemed suitable for publication in PLOS ONE. Congratulations! Your manuscript is now with our production department. 

Kind regards, 

on behalf of

Dr. Kei Masani 

Academic Editor

PLOS ONE